# Adapting Medical Foundation Models for Coronary Artery Calcium Segmentation from CT Imaging

**Jie-En Tsai**                                              JOAN040802@GMAIL.COM
**Po-Chih Kuo**                                              KUOPC@CS.NTHU.EDU.TW
*National Tsing Hua University, Hsinchu, Taiwan*

## Abstract

Automated coronary artery calcium (CAC) segmentation plays an important role in coronary artery disease risk stratification; however, conventional task-specific deep learning models often rely on extensive expert-level annotations to achieve high robustness, which imposes a significant time and resource burden. In this study, we address this limitation by fine-tuning a medical imaging foundation model for CAC segmentation and evaluating its performance under progressively reduced training data. The proposed approach outperforms existing methods when trained on the full dataset and remains competitive even with only 50% of the training data. Moreover, it exhibits smaller performance disparities across sex, age, and CAC severity subgroups compared to a U-Net baseline, highlighting the potential of foundation model fine-tuning for robust and equitable clinical AI applications.

**Keywords:** Foundation Model, Segmentation, Coronary Artery Calcium, Fairness

## 1. Introduction

Coronary artery calcium (CAC) is a critical indicator for assessing a patient's risk of coronary artery disease. Accurate measurement of the CAC score is essential for diagnosis and risk stratification (Sharma et al., 2010). Researchers have explored deep learning (DL) approaches to automate CAC scoring (Yee, 2024; Alirr and Khalifa, 2025). Foundation models offer an effective alternative; trained on large-scale and diverse datasets, they can be adapted to downstream tasks (He et al., 2024) with substantially less training data and achieve superior performance to that of task-specific models (Pai et al., 2024).

Beyond prediction performance, ensuring model fairness across patient subgroups is equally important. Studies have shown that AI models trained on datasets with imbalanced demographic distribution tend to exhibit systematic performance disparities against underrepresented patients across demographic subgroups, potentially leading to insufficient care, delayed diagnosis, and violations of bioethical principles (He et al., 2024; Ricci Lara et al., 2022).

In this work, we adapt BiomedParse, a foundation model developed by Microsoft (Zhao et al., 2025), to the CAC segmentation task via transfer learning on a relatively small training dataset. Our objective is to investigate whether a foundation model can match or outperform conventional task-specific models under reduced data requirements, and evaluate model fairness across clinical and demographic subgroups to assess the equitability of the fine-tuned foundation model.

## 2. Method

**Datasets.** We use two datasets. The COCA dataset (Stanford University and Stanford AIMI, 2021) contains both gated and non-gated CT images; 437 patients of gated images were used after manual data cleaning. The second dataset is a non-contrast cardiac CT images dataset (Kazemi et al., 2023), which contains 120 subjects (43 patients, 77 healthy people), along with corresponding demographic and CAC-related annotations. We utilized this dataset to analyze model fairness across demographic and clinical subgroups based on the ground truth annotations.

**BiomedParse.** BiomedParse (Zhao et al., 2025) is a SEEM-based (Zou et al., 2023) foundation model that supports segmentation, detection, and recognition of biomedical objects across nine imaging modalities via text prompts within a single unified framework. We fine-tune BiomedParse to investigate whether the foundation model can achieve performance comparable or superior to that of task-specific models under data-limited conditions.

**Phase 1: Low-data fine-tuning.** We evaluated performance across four proportions (100%, 75%, 50%, and 25%) using a five-fold cross validation strategy. For each proportion, the held-out test set remained fully intact while only the training folds were downsampled to the target proportion. Each model was trained for 10 epochs, and the final performance was reported as the average across the five folds.

**Phase 2: Subgroup fairness evaluation.** We used the non-contrast cardiac CT dataset (Kazemi et al., 2023) and evaluated model performance with respect to sex (male, female), CAC score (0, 1–99, 100–299, and $\geq$300) according to the CAC-DRS classification (Hecht et al., 2018), epicardial tissue volume (quartile-based), and age (0-50, 51-65, >65 years) to reflect different cardiovascular risk stages while maintaining balanced subgroup sizes. To further assess equitability, we adapted the framework (Seyyed-Kalantari et al., 2020), replacing TPR with MAE, RMSE, and bias as performance metrics. For each performance metric M, the performance gap is defined as follows. For binary attributes (i.e., sex), the gap is computed as the difference between the two subgroups: $\text{Gap}_g = M_g - M_{\sim g}$. For non-binary attributes (i.e., CAC score, epicardial tissue volume, age), the gap for each subgroup $S_j$ is defined as its deviation from the median performance across all subgroups: $\text{Gap}_{S_j} = M_{S_j} - \text{Median}(M_{S_1}, \ldots, M_{S_k})$.

## 3. Results and Discussion

We compared our fine-tuned BiomedParse with a U-Net baseline trained in this study using the same five-fold cross-validation protocol, as well as two Attention U-Net-based approaches (Yee, 2024; Kazemzadeh et al., 2021), and the anatomically guided cascaded U-Net framework with vessel prior (Alirr and Khalifa, 2025), all trained on the COCA dataset (Stanford University and Stanford AIMI, 2021).

When trained on the full dataset, our fine-tuned BiomedParse achieved the highest Dice score among the compared methods, and it slightly outperformed the conventional task-specific approaches and achieved performance comparable to the cascaded framework incorporating vessel priors (Alirr and Khalifa, 2025) with only 50% of the available data.

Regarding the fairness analysis, the fine-tuned BiomedParse generally demonstrates smaller performance gap than the U-Net baseline in sex, age, and CAC score attributes for all three metrics (MAE, RMSE, and bias), suggesting improved equitability across these

clinical subgroups. In contrast, the result for the epicardial tissue volume attribute shows less clear improvement. Both models exhibit subgroup-specific bias across epicardial tissue volume attribute, with the fine-tuned BiomedParse showing more negative bias in the >122 ml group, and the U-Net baseline in the ≤70 ml group.

Table 1: Comparison of Dice scores between the proposed method and prior works for coronary artery calcification segmentation.

| Methods | Dice |
| --- | --- |
| U-Net baseline | 0.817 |
| Attention U-Net + Focal Loss (Kazemzadeh et al., 2021) | 0.739 |
| Attention U-Net + Gaussian Blur (Yee, 2024) | 0.84 |
| Selective U-Net Ensemble (Alirr and Khalifa, 2025) | 0.843 |
| Selective U-Net Ensemble with Vessel Prior (Alirr and Khalifa, 2025) | 0.855 |
| **This work (100% training data)** | **0.857** |
| **This work (75% training data)** | **0.852** |
| **This work (50% training data)** | **0.848** |
| **This work (25% training data)** | **0.827** |

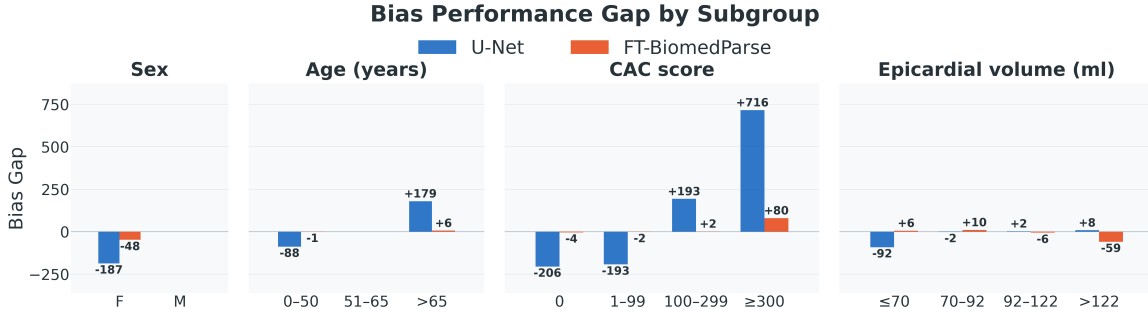

Figure 1: Bias performance gap comparison between U-Net and fine-tuned BiomedParse across subgroups.

## 4. Conclusion

We fine-tune BiomedParse on the CAC segmentation task and demonstrates that the fine-tuned foundation model achieves performance comparable or superior to conventional task-specific segmentation models while requiring substantially less training data. Notably, the model maintains stable performance even when trained on only 50% of the dataset.

Compared with the U-Net baseline, the fine-tuned model shows reduced performance disparities across most clinical subgroups, suggesting improved equitability. However, systematic bias cannot be fully eliminated, which may influence downstream risk stratification and clinical decision-making. Future work should examine how adjusting training data composition can reduce bias and improve model performance across subgroups.

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
