# OpenReview forum: "Adapting Medical Foundation Models for Coronary Artery Calcium Segmentation from CT Imaging"
_MIDL.io/2026/Short_Papers — MIDL 2026 - Short Papers Poster_

### Official Review · Reviewer_wqsg · 2026-05-03
**Fine-tuning a foundation model for calcium scoring**

**Rating:** 3
**Confidence:** 5

**Review:**

The paper is generally well-written and easy to follow. Results are clear and realistic. My main concerns are with the motivation of the work and its validation.

Firstly, the statement that ‘however, conventional task-specific deep learning models typically require large annotated datasets that are challenging to obtain clinical settings.’ Is not entirely correct when it comes to calcium scoring. There are large-scale public datasets, and calcium scoring is not particularly challenging, so a large private dataset is relatively easy to obtain.

Second, the work is compared to several baseline models. However, while the foundation model is fine-tuned on 25-50-75-100% of the data, the baseline models are trained only on 100% of the data. This makes it difficult to assess the extent to which the data efficiency in foundation model finetuning is unique. Moreover, it's unclear what kind of architecture is used in BioMedParse. Is this also a U-Net? To which of the baseline models in Table 1 should we best compare the foundation model, architecture-wise?

**Summary:**

The paper studies the use of a pretrained foundation model (BioMedParse) for segmenting coronary artery calcifications in cardiac CT. Results show that this model can be fine-tuned for calcium scoring, with segmentation performance exceeding that of a baseline U-Net model, even when trained on a subset of the data.

**Strengths:**

- The paper looks at equitability across different subpopulations and finds that this is improved for the pretrained model.
- The paper shows strong results for a fine-tuned foundation model for calcium scoring.

**Weaknesses:**

- The motivation of the work is weak. It’s hard to argue that calcium scoring requires the use of pretrained foundation models, as large-scale public data sets are available (e.g. COCA, which the authors even use).
- The results and discussion do not sufficiently disentangle the effect of (1) data set size, (2) architecture, and (3) pre-training and fine-tuning strategy,.

**Justification Of Rating:**

Generally interesting paper with strong results, but the motivation and discussion lack depth.

---

### Decision · Program_Chairs · 2026-05-08

Accept (Poster)